# How Does Protein Nutrition Affect the Epigenetic Changes in Pig? A Review

**DOI:** 10.3390/ani11020544

**Published:** 2021-02-19

**Authors:** Pablo Jesús Marín-García, Lola Llobat

**Affiliations:** 1Departamento Producción y Sanidad Animal, Salud Pública y Ciencia y Tecnología de los Alimentos, Facultad de Veterinaria, Universidad Cardenal Herrera-CEU, CEU Universities, 46010 Valencia, Spain; pablo.maringarcia@uchceu.es; 2Grupo de Fisiopatología de la Reproducción, Departamento Producción y Sanidad Animal, Salud Pública y Ciencia y Tecnología de los Alimentos, Facultad de Veterinaria, Universidad Cardenal Herrera-CEU, CEU Universities, 46010 Valencia, Spain

**Keywords:** amino acids, epigenetic changes, lysine, methyl group donors, methionine, ideal protein, precision nutrition, pig

## Abstract

**Simple Summary:**

Epigenetic mechanisms regulate gene expression and depend of nutrition. In farm animals, and concretely, in pigs, some papers on protein nutrition have been realized to improve several productive traits. Changes in protein diet influence on epigenetic mechanisms that could affect productive and reproductive traits in individuals and their offspring. The purpose of this review was to update the current knowledge about the effects of these nutritional changes on epigenetic mechanisms in pigs.

**Abstract:**

Epigenetic changes regulate gene expression and depend of external factors, such as environment and nutrition. In pigs, several studies on protein nutrition have been performed to improve productive and reproductive traits. Indeed, these studies aimed not only to determine broad protein requirements but also pigs’ essential amino acids requirements. Moreover, recent studies tried to determine these nutritional requirements for each individual, which is known as protein precision nutrition. However, nutritional changes could affect different epigenetic mechanisms, modifying metabolic pathways both in a given individual and its offspring. Modifications in protein nutrition, such as change in the amino acid profile, increase or decrease in protein levels, or the addition of metabolites that condition protein requirements, could affect the regulation of some genes, such as myostatin, insulin growth factor, or genes controlling cholesterol and glucose metabolism pathways. This review summarizes the impact of most common protein nutritional strategies on epigenetic changes and describes their effects on regulation of gene expression in pigs. In a context where animal nutrition is shifting towards precision protein nutrition (PPN), further studies evaluating the effects of PPN on animal epigenetic are necessary.

## 1. Introduction

The first definition of epigenetics was done in 1940 by Waddington [1]. Currently, different epigenetic regulation processes are known. The main acquaintances are DNA methylation, histone modifications, regulation by non-coding RNAs, such as microRNAs (miRNAs), and mechanisms that control chromatin organization. These processes can regulate gene expression independently of DNA sequence [2]. The first documented epigenetic modification was DNA methylation, and it was discovered in 1975, related to X-chromosome inactivation [3]. Lately, histone modifications and the role of chromatic structure in the regulation of the genome were elucidated [4]. The latest epigenetic regulation process discovered has been the non-coding RNA [5]. The influence of the environment on epigenetic changes has been demonstrated in plants and animals. Nutritional factors, such as levels of protein, and methyl group donors (some essential amino acids and other metabolites related to them) could also influence these epigenetic regulation [6,7]. In farm animals, different nutritional strategies are being investigated specifically in protein and amino acids content, which are directly related to both productive traits, as reduction of boar taint in meat, and to improve the meat quality [8,9]. These changes could have an epigenetic effect on animals, and their offspring. In this review, we summarize different strategies in pig’s nutrition, and their possible consequences in epigenetic regulation.

## 2. Nutrition in Pigs

Protein nutrition is essential in pig production, and it is being widely studied. Protein and amino acids have several functions (structural, regulatory, transporter, defensive, enzymatic, or even contractile) for animals, and they direct almost all life processes. Proteins are macro-biomolecules consisting of one or more amino acids chains, and they are synthesized from them, which become available either from the end products of digestion [10]. There exist two types of amino acids: those that the animals can synthesize (non-essential) and those that cannot synthesize by themselves (essential), and whether an amino acid is essential or not always depend on the species. Concretely, essential amino acids in pigs are methionine, lysine, histidine, isoleucine, leucine, phenylalanine, threonine, tryptophan, and valine, and they are necessary to control and provide by the diet, either though raw materials, or in some cases synthetically, due to the high nutritional requirements in livestock species.

Although protein and amino acids are considered nutrients (those components capable of being utilized by animals), some authors considered as additives, i.e., substances which, when incorporated in feeding-stuffs, are likely to affect their characteristics or livestock production [11] when they are added in a synthetic way. In this review, we refer to the presence of the nutrient regardless of whether it has been supplied by raw materials or synthetically.

### 2.1. Quality and Quantity of Protein

For the correct evaluation of the epigenetic effects of some nutrients, it is necessary to know the quality and quantity of them, and their interaction. In general terms, animals can be fed following two strategies, ad libitum or restricted. Regarding ad libitum system, the factor that will most affect the total amount of nutrients ingested will be the capacity of feed intake. Feed intake has been extensively studied both in growing animals [12], as well as reproductive animals [13,14], and it depends on many factors, like sex, breed, physiological state, environmental conditions, and housing system, but above all, it is also dependent of the quality of the diet [15,16,17]. Nevertheless, life weight is the parameter that reflects feed intake. There are certain physiological states (mainly pregnancy), when the animals are subjected to a food restriction, and this has direct effects on the total ingestion of protein and amino acids.

Regardless the nutrient quality, from the total crude protein (CP) ingested only a part, that is digested by enzymatic action and converted to amino acids and peptides [10], is absorbed into the stomach and, mainly, the small intestine [18]. This proportion corresponds with ileal digestible CP. The rest (ileal indigestible CP) goes to the large intestine, where it could be used or modified by the microbiota action (fecal digestible CP), or eliminated in feces (fecal indigestible CP). Digestible amino acids requirements could be calculated by either considering (real) or not (apparent) the endogenous flow (only the digestible fraction is capable of creating these epigenetic effects). For this reason, the formulation according to the digestible requirements at the true ileal levels is recommended; this is close to the ideal protein concept that refers to a situation where all essential amino acids are collimating for performance, so that the amino acids supply exactly matches the amino acids requirement [19]. In summary, a correct adjustment will lead to a better productivity, less pollution, and better health, thus improving the welfare state of the animals.

For the aforementioned, a large number of works to find the ideal protein have been performed in some productive species [20,21,22], including pigs. Usually, amino acid requirements have been determined by using dose–response methods which show the response of the performance to ascending contents of the amino acid studies [23]. However, there exist other techniques, such as controlling the amount of amino acid (in blood, plasma or serum), or even plasmatic urea nitrogen, which corresponds to the amount of nitrogen in form or urea circulating in the bloodstream [24]; it could be related with performance in pigs [25,26]. Nitrogen from unused amino acids presents an energy cost for the animal—three ATP molecules for every nitrogen molecule excreted [27]—related to the urea cycle. For this reason, not taking advantage of protein has a negative impact on two major aspects, as more energy used and more protein excreted. Furthermore, certain nutrients affect the requirements of another. In fact, methionine supplies methyl groups, and they are used in several metabolic pathways. These methyl groups also can be donated by choline, but both are eventually converted to betaine throughout these metabolic pathways. For this reason, an increase in both (choline and betaine) could minimize the use of methionine by these metabolic pathways, and therefore, its nutritional requirements [28,29].

### 2.2. Ideal-Protein Concept

Formulation according ideal protein is possible only if accurate knowledge exists about the requirements for all amino acids [19]. Currently, the global requirements for animals are widely studied, and they are collected in Figure 1, where current recommendations of protein according different physiological phases of pig production are shown [30,31,32]. These values are well-established for the medium of animals, but obviously they are not good for all animals. To be able to formulate into ideal protein, it is necessary to know the optimal nutritional requirements for each animal type, and there are several aspects that affect nutritional requirements, such feed efficiency, breed, environment, and performance, among others [33]. For this reason, there are many trials that try to relate the essential amino acid profile in the optimization both productive and reproductive traits, and some are developed below. Some works have studied the effects of lysine [25,33,34], methionine [35], arginine, and other essential amino acids [35,36,37,38,39,40] in several productive traits, like life weight, feed intake, feed conversion ratio, retention, or digestion. These results are explained by the greater availability of both protein and energy that allows the animals to optimize their productive traits. The different levels of amino acids not only affect growing pigs; this profile affects reproductive traits in pregnancy and lactating sows, uterine regression, milk and colostrum production, placenta and membrane functions, fetus growing, total piglets born, and weaned pigs, among others. Some works have studied the effects of lysine [23,41,42,43], methionine [44,45], arginine, valine, threonine, and tryptophan [26,43,46] in all parameters described above.

Several studies have been carried out in order to determine the optimal levels of amino acids for each stage in the life of the animal, which is leading towards the concept of precision protein nutrition (PPN). This PPN would try to assess the needs of the main amino acids in real time [47]. Thus, this new approach to animal nutrition is completely dynamic and requires continuous changes in the diet, to adapt to each situation [36]. This solution could minimize the epigenetic effects of an excess or lack of protein (or a certain amino acids) in animals, and specifically in pigs.

## 3. Epigenetic Modifications

### 3.1. DNA Methylation

DNA methylation is an essential process to regulate some biological functions in mammals, such as chromosomal stability, genomic imprinting, and X-chromosome inactivation. It occurs when methyl groups are added to the cytosine of CpG islands (Figure 2) [48]. This addition is catalyzed of three conserved enzyme families in mammals, DNA methyltransferase 1 (DNMT1), DNMT3a, and DNMT3b, which are essential to mammalian development and biological functions [49,50,51]. While DNMT1 maintains the methylation patterns during DNA replication, DNMT3a and DNMT3b are de novo methyltransferases [52]. The DNMT1 protein is expressed in somatic tissues and proliferating cells, and several isoforms have been detected [53,54]. In the case of DNMT3a and DNMT3b, alternative transcripts have been detected in the human and mouse [55].

To realize their function, these enzymes need a methyl donor, so they are directly influenced by methyl groups derived from the diet, such as choline/betaine, methyl-folate, or methionine, that are precursors to the universal methyl-donor S-adenosylmethionine (SAM) [56,57,58,59]. An example is the regulation of insulin-like growth factor II (IGF2), which is negatively regulated by a H19 gene, that regulation, in turn, depends of concrete region methylation, Igf2DMR2. Pregnant rats with a choline-deficient diet presented hypermethylation of these regions, so H19 is inhibited and IGF2 expression increases [60]. Epigenetic regulation of this gene and its relationship with nutrition have been well-known since the Dutch Hunger Winter in 1944/1945 in humans, when pregnant women exposed to famine had descendants with less DNA methylation at the IGF2 gene, and these losses were maintained up to 60 years [61]. In livestock, different authors indicate the relevance of nutrition in epigenetic marks. Murdoch et al. (2016) published an extensive review about the methylation effect of quality and quantity of nutrients in different livestock species [62]. The most important nutrient related to epigenetic effects and, concretely, with effect in DNA methylation, are methyl group donors, such as methionine or another methyl group donor used as an alternative to methionine (mainly, folate and betaine). Betaine injections in eggs increase DNA methylation, regulating hepatic cholesterol metabolism in chicks [63]. Moreover, pregnant sows that were fed with betaine-supplemented diets presented modifications of methylation patterns in newborn piglets [64]. In sheep, deficient diets in vitamin B12, folate, and methionine provoke hypomethylation in offspring that relates to birth weight, alterations in immune responses, and elevation of blood pressure [61].

Quantitative trait loci (QTL) have been found in pig for most economically important traits, such as carcass and meat quality [65,66,67]. Some of these QTL are regulated by methylation [68,69] or another epigenetic mechanism [70] and could affect phenotypic variation in livestock production. Therefore, studies on epigenetic regulation, and specifically through DNA methylation of characters of interest for livestock production, would be necessary, mainly from the point of view of how the nutrition of animals affects these methylation changes.

### 3.2. Histone Modifications

Since 1964, the post-translational modifications of histones have been well-known [71]. Acetylation, phosphorylation, and methylation of histones could affect chromatin structure and influence transcription and gene expression (Figure 3) [72]. Modifications in histone 1 (H1) (linker histone), H2A, H2B, H3, and H4 core histone families are related to different functions, such as chromatin dynamics, transcriptional activation, heterochromatin formation, telomere stabilization, or gene silencing, among others. These functions could regulate gene expression and develop as different genetic disorders or early development in mammals [73,74,75].

Acetylation occurs in lysine residues of histones, and it regulates by two families of enzymes, histone acetyltransferases (HATs) and histone deacetylases (SDCAs). The HATs catalyze the transfer of an acetyl group of lysine side chain, whereas HDAC enzymes catalyze the reverse process [76]. Histone phosphorylation takes place on serine, threonine, and tyrosine and is regulated by kinases and phosphatases [77]. Finally, histone methylation is carried out on the side chain of lysine and arginine, with specific methylases and demethylases [78,79]. Histone modifications are related to the regulation of multiple biological process. One of the most important process regulated by histone modifications is the early mammalian development. One example is the formation of the trophectoderm and the inner cell mass that is related to modifications in histone 3, concretely in the 4 and 27 lysine residues (H3K4 and H3K27) in mouse embryos [80]. Histone modifications are necessary for more complex regulatory mechanisms in vivo. Indeed, histone methylation is related to genomic imprinting, HOX gene expression, and the regulation of pluripotency in mammalian early development [81].

Some studies indicate the relationship between different nutrients and histone modifications. Regarding that, one of the most studied nutrients are vitamins. The availability of retinoic acid (RA), a natural derivative of vitamin A, modifies the histone patterns. RA treatment increases H3K4 and decreases H3K27, and it increases H3K9 and H3K14 acetylation in humans [82]. A diet deficient of vitamin A decreases HAT activity and H3 and H4 histones acetylation in rats [83]. Another vitamin that affects histone modifications is vitamin C, which is also linked with the reduction of H9K9 dimethylation in mouse embryonic stem cells [84]. Other compounds of diets related to histone modifications are organosulfur compounds, sodium butyrate, genistein, bioflavonoids, or catechins [85]. Studies related to effect of vitamins or other nutrients in farm animals are not carried out, except related to protein levels. Deficient-protein diets improve H3 acetylation in the promotor of myostatin gene that produces alterations in gene expression in pigs [86,87].

### 3.3. Small Non-Coding RNA: miRNAs as Epigenetic Regulators

In 1993, the first miRNA lin-4 was discovered in the nematode Caenorhabditis elegans, but its importance was only understood after the discovery of a second miRNA, let-7 [88,89]. MiRNAs are small non-coding RNA molecules of approximately 20–24 pairs in length, are therefore they are unable to code for proteins [90,91,92]. They are usually phylogenetically conserved and have been shown to play a crucial role in the regulation of gene expression and cellular processes [92,93]. Since their discovery, thousands of miRNAs have been identified, and their presence or absence regulates different functions in different organisms. The biogenesis of miRNAs is a complex process that requires two endonucleases, Drosha and Dices, and the nuclear transport receptor Exportin 5 (XPO5) (Figure 4) [94]. The miRNAs, along with other less-known non-coding RNAs, as piwi-interacting RNA (piRNA), small-interfering RNA (siRNA), and small nucleolar RNA (snoRNA), regulate gene expression in some mechanism, including heterochromatin formation and inhibition of translation [95,96,97].

Like other epigenetic regulators, miRNAs could also be affected by nutrients. Expression of miRNAs could also be influenced by vitamins, so RA regulates the miRNAs expression in different biological tissues, such as embryos, the heart, muscles, or nervous tissue [99]. Moreover, low-protein diets provoke the reduction of miR-136 and miR-500, affecting myostatin gene expression in pigs [87].

### 3.4. Other Non-Coding RNAs as Epigenetic Regulators

Additionally, other non-coding RNAs appear to act as epigenetic regulators. For instance, there is a growing body of evidence suggesting that small interfering RNAs (siRNAs), small nucleolar RNAs (snoRNAs), circular RNAs (circRNAs), piwi-interacting RNAs (piRNAs), or long non-coding (lncRNAs) have a pivotal role in the epigenetic landscape (reviewed elsewhere [100,101,102]). However, the direct relationship between these non-coding RNAs and nutrition still remains unknown. Table 1 summarizes different non-coding RNAs and their impact in epigenetics.

## 4. Epigenetic Effect of Diets in Pigs

### 4.1. Effect of Protein Levels

Different studies have shown the effects of high or low amounts of protein in the diet, on the regulation of gene expression. The first trials indicated an increase in the mRNA expression of renal renin when rats were fed a high-protein diet, which can lead to kidney diseases [113,114]. A similar effect has been detected in liver, where an increase of protein in the diet provokes cellular retinol binding protein I (CRPB I) overexpression [115]. However, not only the protein quantity affects the regulation of genes, but the quality of this protein also alters expression profiles. Endo et al. (2002) observed different expression of 111 genes in rats fed with casein than rats fed with wheat gluten, and drastic changes in expression of DNA-binding inhibitor protein (involved in the regulation of multiple genes) were detected [116]. These results could have a relationship with the different digestibility of these two protein sources [117]. On the other hand, high protein levels increase the gene expression of enzyme serine dehydratase in the liver, related to gluconeogenesis, and causing weight gain in these animals probably [118]. Other studies have analyzed the effect of different amounts of protein in the diet of pregnant females. The results indicated that low protein levels increase the expression of genes related to p53 pathways and negative regulators of cell growth and metabolism, in addition to genes involved in epigenetic changes [119]. These changes in DNA methylation due to a low protein diet seem to be able to be alleviated with a supplementation in methyl group donors, such as folate [120].

The effect of changes of protein diet have also been seen in pigs. For example, in sows, low levels of protein resulted in an increase in myostatin gene expression at weaning; however, the authors did not find differential expression of the 12 miRNAs analyzed, related to the regulation of the expression of this gene [87]. A decrease in the expression of miRNAs as ssc-miR-136 and ssc-miR-500 has been demonstrated, but the regulatory path is not clear yet [87]. Evidence that the level of protein in the diet affects epigenetic regulation has been demonstrated in different studies in pigs. Altmann et al. (2012) observed a different expression of demethylation enzymes DNMT1, DNMT3a, and DNMT3b, with high and low protein diets during porcine pregnancy in skeletal muscle and liver expression patterns [121]. In a posterior study, same authors detected different methylation patterns in metabolic genes with high and lower levels of protein in the diet of pregnancy sows [122], which shows that the amount of protein ingested by the mother during pregnancy has a direct effect on methylation patterns and, therefore, on the epigenetic regulation of the offspring. In accordance with that, low levels of protein in pregnant sows give rise to hypomethylation of the promotor of 3-hydroxy-3-methylgluratyl-CoA reductase (HMGCR) gene (related to cholesterol biosynthesis), and with the decrease of H3 histone monomethylation, lysine 9 of H3 histone monomethylation, and lysine 27 of H3 histone trimethylation, as well as the increase of H3 acetylation, among other epigenetic changes [123]. Besides these effects on cholesterol metabolism and its epigenetic regulation, low protein levels influence the regulation of the insulin growth factor system, decreasing IGF-I levels in liver by the regulation of rapamycin (mTOR) and peroxisome proliferator–activated receptor γ (PPAR γ) interaction [124]. Thus, pregnant sows fed with low- and high-protein diets during pregnancy presented changes in protein and fats metabolism, causing a delay in fetal growth and low birth weights of the offspring [125]. Low levels of protein affect the formation of prenatal myofibers, and they reduce the expression of IGF2 and change the quality of the carcass [126]. These results indicate that a protein imbalance during pregnancy alters the regulation of these metabolic pathways in sows and their offspring, and a deficiency in protein in the maternal diet alters meat quality. Some authors attribute these changes to the imbalance of some micronutrients, as certain essential amino acids, or molecules, such as methyl group donors. For this reason, in the following sections of this review, these metabolites are analyzed individually.

### 4.2. Effect of Methyl Group Donors

Some amino acids, mainly methionine, participate in the in one-carbon-unit metabolism that regulates the synthesis of purines and methylation. Other metabolites, such as betaine, choline, and folate, are used to provide methyl donor for methyltransferases, and their use is related with amino acids requirements. All of them have an important role in the regulation of SAM (Figure 5) [127].

Regarding these methyl group donors, one of the most studied in pregnant sows has been betaine. Betaine is a substrate for the formation of methionine, and further studies indicate that betaine supplementation may modulate gene expression by DNA and histone methylation in human [128,129]. In pigs, the Zhao research group has carried out several studies, in which the epigenetic influence of betaine supplementation in the diet of pregnant sows has been demonstrated. Concretely, this supplementation increases liver cholesterol, changes the expression of different genes related to gluconeogenesis metabolism, and inhibits liver cells proliferation. For example, IGF2 gene affects gluconeogenesis, and its expression in the hippocampus of newborn piglets is controlled by different epigenetic mechanisms, such as DNA methylation, histone modifications, and expression or repression of some miRNAs [64,130,131,132]. The effects of a deficiency in folate and their relationship between epigenetic regulation have been extensively studied in humans [133]. In pigs, studies are not as extensive; however, a similar effect of betaine on DNA methylation in piglets has been demonstrated [134]. Recently, He et al. (2020) studied the effect of methyl donors, such as folate, choline, and methionine supplementation in pregnant sows, and the results showed an increase of proteins of one-carbon-unit metabolism in females and piglets, and skeletal muscle differentiation increase in piglets, probably regulated by epigenetic modifications [135]. However, since, in this study, the supplementation was performed with all the methyl group donors simultaneously, the effect of each of them in pigs remains to be discovered.

Some studies have been realized in other mammalian species and demonstrated the beneficial effect of methionine supplementation in epigenetic mechanisms. On one hand, methionine supplementation in rats is capable of inhibiting the expression of DNMT1 and histone methylation, thereby preventing epigenetic alterations induced by diabetes [136]. On the other hand, the methionine supplementation in the diet of pregnant female cattle increases DNMT3a, whereas global DNA methylation is lower in placenta [137]. Considering that the methylation state fluctuates throughout the sow’s reproductive cycle [138], studies that analyze the effect of methionine supplementation in this species are necessary, and they could have more relationship about an imbalance in the quality of the protein (essential amino acid profile).

### 4.3. Effect of Lysine

Lysine is one of the essential amino acids in pigs, and the requirements are higher than other essential amino acids. Regarding epigenetic mechanisms, both histone acetylation and histone methylation occurs in lysine residues of histones [76,79]. Methylation of lysine residues of non-histones protein has been discovered with another mechanism of post-translational regulation that regulates chromatin remodeling, gene transcription, protein synthesis, signal transduction, and DNA repair [139].

Lysine deficiency in pigs affects different productive traits. Low levels of lysine in pig diets decrease the average daily gain and loin percentage, and increase feed-to-gain conversion ratio, back-fat thickness, and intramuscular fat content [140,141]. The molecular mechanisms that produce these effects are unknown; however, some studies have shown that diets with a low lysine content decrease the expression of some genes, such as genes that code for amino acid transporters and muscle growth, such as myosin [142,143]. Low levels of lysine cause a reduction in IGF-1 plasmatic levels, without decreasing the levels of its mRNA in the liver, suggesting a post-transcriptional regulation [144]. Although these findings seem to point to an influence of lysine deficiency on gene regulation, either pre- or post-translational, more studies will be necessary to find out the regulatory processes that underlie these changes in production parameters.

About the effects of lysine excess, most authors agree on the low toxicity of lysine, mainly due to a slow entry of it into the bloodstream, good elimination through urine, longer time for release of lysine from circulation to muscle during storage, and liver metabolism [145]. Nonetheless, hyperlysinemia has been related to mental motor retardation, seizures, muscle hypotonia, spasticity, abdominal cramps, and diarrhea in humans [146,147]. Studies in pigs, in this regard, are scarce. For example, Edmonds and Baker (1987) founded a decrease of daily weight and food intake when animals were fed with higher lysine levels [148]. As with lysine deficiency, the underlying mechanisms are unknown.

Continuing to research the effects of the excess and deficiency of lysine both in production parameters and in mechanisms of epigenetics regulation would be of high interest for the swine industry and for animal nutrition.

## 5. Conclusions

This review analyzed different effects of both protein and amino acid levels in the diet, on different epigenetic regulatory mechanisms, such as DNA methylation, histone modifications, and regulation by non-coding RNAs (miRNA). Given that nutritional requirements are variable for each animal and changing in a dynamic way, new animal nutrition techniques go towards precision protein nutrition. In this context, where amino acid content is variable, it is important to know the effects of how these different levels of proteins and amino acids could modify these epigenetic changes, since these interactions not only affect the productive traits of the animal, but also other traits both of its own and of its offspring.

## Figures and Tables

**Figure 1 animals-11-00544-f001:**
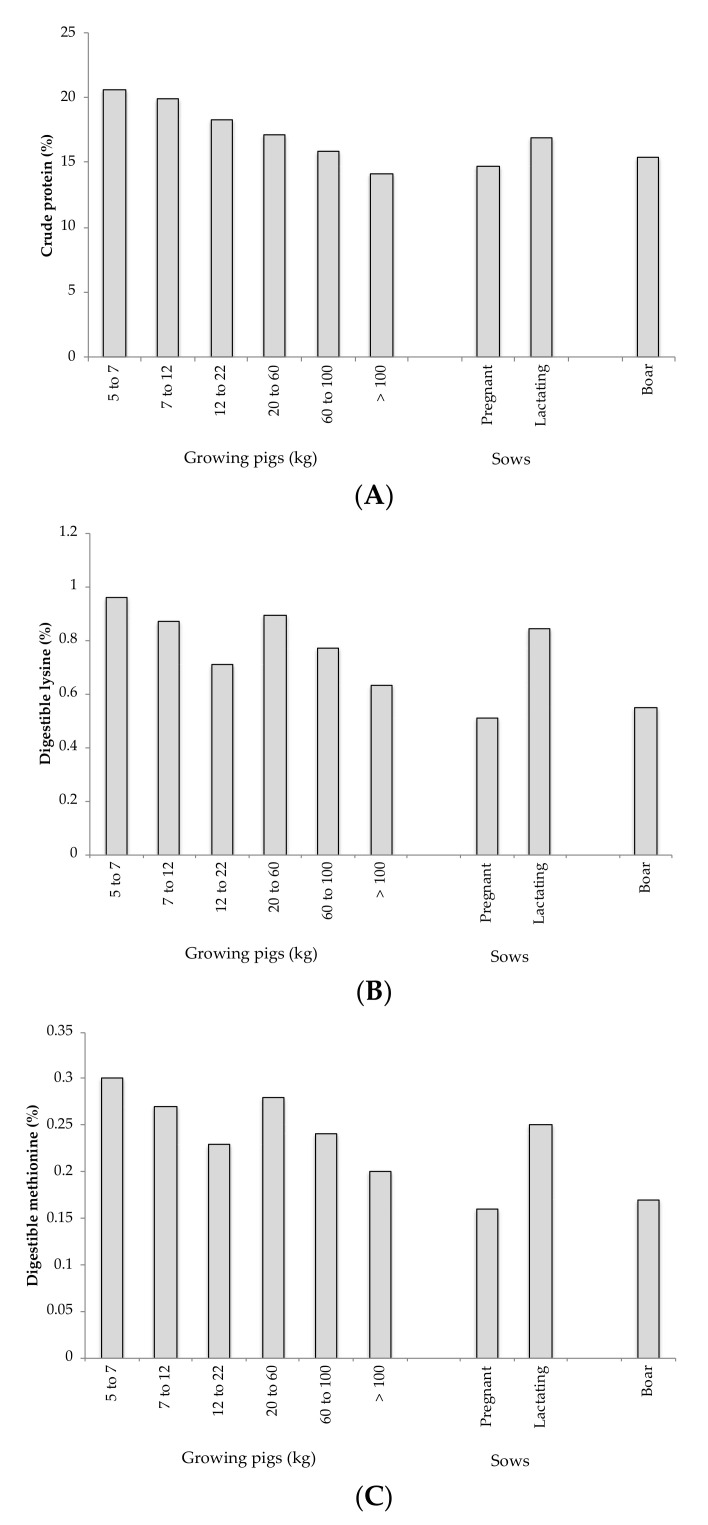
Current recommendations of crude protein (**A**), digestible lysine (**B**), and digestible methionine (**C**) of the different physiological phases of pig production [30,31,32].

**Figure 2 animals-11-00544-f002:**
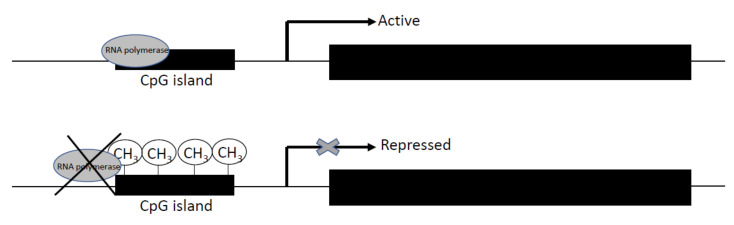
Scheme of the regulation of gene expression by DNA methylation. Methyl groups were added by DMNT1, DNMT3a, and DNMT3b enzymes in cytosine of CpG island. As result, the RNA polymerase is unable to bind to DNA sequence, and the gene promoter is not expressed.

**Figure 3 animals-11-00544-f003:**
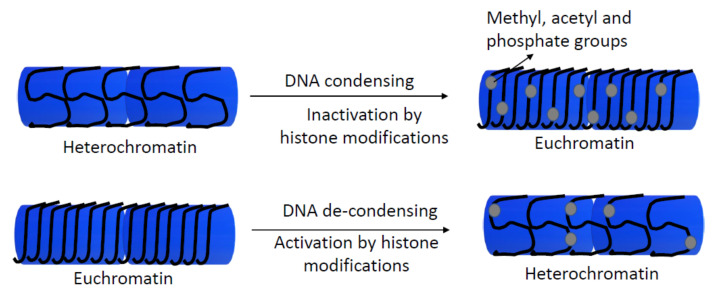
Scheme of the regulation of gene expression by histone modifications. Enzymes as histone methyltransferase and acetyltransferase add or remove methyl, acetyl, and phosphate groups of histones, influencing to transcriptional machinery, and they activate or repress gene expression by de-condensing or condensing chromatin, respectively.

**Figure 4 animals-11-00544-f004:**
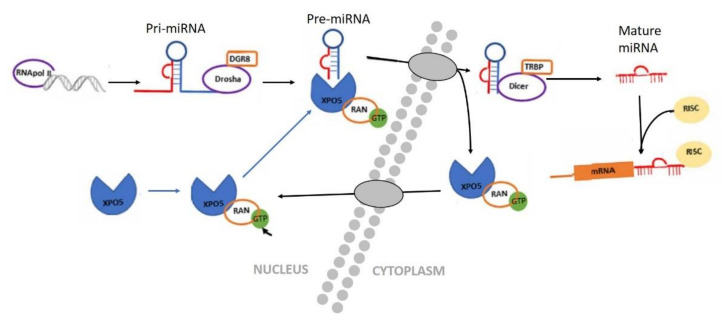
Diagram of miRNA biogenesis. In the nucleus, miRNA that are transcribed by RNA polymerase II or III as primary transcripts (pri-miRNA). The enzyme Drosha and its cofactor DGR8/Pasha then cut the pri-miRNA molecule, to form a pre-miRNA. This pre-miRNA is actively transported from the nucleus to the cytoplasm by the nuclear transport receptor Exportin 5 (XPO5) in a Ran-GTP-protein-dependent manner by XPO5. In the cytoplasm, the pre-miRNA is cut by a second enzyme, called Dicer, until the formation of a mature and short double-stranded miRNA molecule. The miRNA duplex unwinds and is incorporated into the RNA induced silencing complex (RISC) protein [98].

**Figure 5 animals-11-00544-f005:**
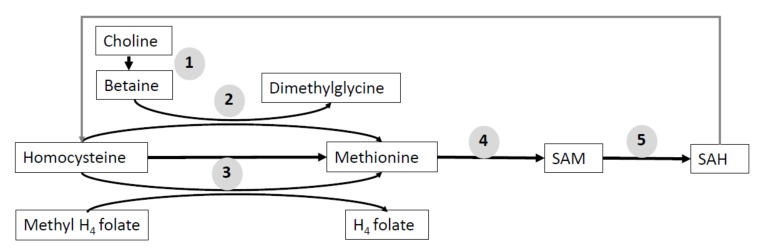
Schematic explanation of one-carbon unit metabolism. Methyl donors participate in the transfer of one-carbon units for the methylation reactions. The enzymes related with pathways are (**1**) choline dehydrogenase, (**2**) betaine homocysteine methyltransferase, (**3**) methionine synthase, (**4**) S-adenosylmethionine synthase, and (**5**) S-adenosylmethionine synthase. SAH, S-adenosylhomocysteine; SAM, S-adenosylmethionine.

**Table 1 animals-11-00544-t001:** Different non-coding RNAs related to epigenetics.

Non-Coding RNA	Functions	References
Small Interfering RNA (siRNA)	Gene silencing	[103]
Small Nucleolar RNA (snoRNA)	rRNA modifications	[104,105]
Circular RNA (circRNA)	miRNA sponge, regulation of gene transcription, RNA binding protein sponge	[106,107]
Piwi-interacting RNA (piRNA)	Transposon repression, DNA methylation	[108,109]
Long Non-coding RNA (lncRNA)	X-chromosome inactivation, telomere regulation, imprinting	[110,111,112]

## Data Availability

Not applicable.

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
