# Peer review of "How Does Protein Nutrition Affect the Epigenetic Changes in Pig? A Review"

_animals, 2021, doi:10.3390/ani11020544_

Round 1
Reviewer 1 Report
How does protein nutrition affect the epigenetic changes in pig? A review
This manuscript “How does protein nutrition affect the epigenetic changes in pig? A review
“ presents an interesting title, but the content is not rich enough.
- It seems that there are very few studies about the field for the authors to summarize.
- The authors presented miRNAs as one of the epigenetics mechanisms, but it seems the authors could not give enough examples to support the idea.
- The authors could not have enough evidence to explain how epigenetics marks changed due to diet and which mechanism behind it.
- What is a perspective, the manuscript is poorly ended?
- Needed improvement in the writing.
Line 13: realize change to realized. Also, animal change to animals
Line 15: individual to individuals
Line 27:, or cholesterol and glucose metabolism pathways, do the authors mean genes involved in cholesterol and glucose metabolism pathways?
Line 28: describing change to describes.
Line131: Change show to shown.
Line 146-147: Could the authors shortly summarize information for other essential amino acids as well.
Figure 1 is very poor quality, where did the information from the figure come from (common literature or specific sources)?
Line 151: What did the authors mean for all this.
Line 151: change “there are many works that study” to various studies.
Line 151-152: please rewrite the sentences.
Line 155-156: which solution did the authors want to mention, the whole paragraph is very confused.
Line 160: add "an" before essential
Line 179: change gene name to Italic
Line 197 -198: Might better to change the reference to Animal QTL databases as they provided the list of QTLs for each trait. (https://www.animalgenome.org/cgi-bin/QTLdb/SS/index)
Line 249” write the species name in Italics
Sections 3.3: When talking about the epigenetics regulations, I think other groups of ncRNAs also important. As the review manuscript, the authors should provide an overview of ncRNAs before emphasizing the miRNAs roles.
Section 4.4 about pigs, the authors did not need to list many about rats (line 278-296)
Line 297-300: add a reference.
Line 300-301: it is not clear.
Line 303-310: it is not clear as well, that the correlation (significant, negative or positive) the reference found.
Line 319: “These epigenetic changes produce different phenotype in offspring”, it is hard to state that, it is not clear and not true.
Author Response
Dear reviewer,
The authors would like to thank you for the useful feedback on our review.
We have made several editorial changes in response to the critiques and have included these in the revised manuscript.
The following is a point-by-point response with line numbers added to indicate where specific changes have been made. We have included the reviewer comments and requests in black font followed by our responses in blue font.
Line 13: realize change to realized. Also, animal change to animals
Done.
Line 15: individual to individuals
Done.
Line 27:, or cholesterol and glucose metabolism pathways, do the authors mean genes involved in cholesterol and glucose metabolism pathways?
The sentence has been rewritten as “genes controlling cholesterol and glucose metabolism pathways”.
Line 28: describing change to describes.
Done.
Line131: Change show to shown.
Done.
Line 146-147: Could the authors shortly summarize information for other essential amino acids as well.
The sentence has been rewritten as “arginine, valine, threonine, and tryptophan”.
Figure 1 is very poor quality, where did the information from the figure come from (common literature or specific sources)?
Figure 1 has been modified with high quality and the references has been added.
Line 151: What did the authors mean for all this.
“For all this” has been eliminated.
Line 151: change “there are many works that study” to various studies.
Done.
Line 151-152: please rewrite the sentences.
The sentence has been rewritten as “Several studies have been carried out in order to determine the optimal levels of amino acids for each stage in the life of the animal, which is leading towards the concept of precision protein nutrition”.
Line 155-156: which solution did the authors want to mention, the whole paragraph is very confused.
The paragraph has been change as “This PPN would try to assess the needs of the main amino acids in real time [46]. Thus, this new approach to animal nutrition is completely dynamic and requires continuous changes in the diet to adapt to each situation [35]. This solution could minimize the epigenetic effects of an excess or lack of protein (or a certain amino acids) in animals, and specifically, in pigs”.
Line 160: add "an" before essential
Done.
Line 179: change gene name to Italic
Done.
Line 197 -198: Might better to change the reference to Animal QTL databases as they provided the list of QTLs for each trait. (https://www.animalgenome.org/cgi-bin/QTLdb/SS/index)
The reference has been added.
Line 249” write the species name in Italics
Done.
Sections 3.3: When talking about the epigenetics regulations, I think other groups of ncRNAs also important. As the review manuscript, the authors should provide an overview of ncRNAs before emphasizing the miRNAs roles.
The authors fully agree with the reviewer's comment about other types of epigenetic regulation involving other RNAs, such as ncRNAs or lcn-RNAs. However, the bibliography about the relationship between epigenetic regulation by these RNA molecules and protein nutrition is very scarce, so the authors decided not to include them in the manuscript.
Section 4.4 about pigs, the authors did not need to list many about rats (line 278-296)
Some information about rats was added in the manuscript, so the first studies were carried out in this species. Nevertheless, “in rats” has been removed.
Line 297-300: add a reference.
Reference has been added.
Line 300-301: it is not clear.
The sentence has been rewritten as “A decrease in the expression of miRNAs as ssc-miR-136 and ssc-miR-500 has been demonstrated, though the regulatory path is not clear yet”.
Line 303-310: it is not clear as well, that the correlation (significant, negative or positive) the reference found.
The sentence has been rewritten as “Altmann et al. (2012) observed different expression of demethylation enzymes DNMT1, DNMT3a, and DNMT3b, with high and low protein diets during porcine pregnancy in skeletal muscle and liver expression patterns“ to facilitate its understanding.
Line 319: “These epigenetic changes produce different phenotype in offspring”, it is hard to state that, it is not clear and not true.
This sentence has been removed.

Reviewer 2 Report
Presented manuscript is quite interesting from the theoretical and practical point of view.
However, it must be improved prior to publish in Animals journal.
These are some examples:
- sentences are too long and sometimes hard to understand
- line 21 - tried
- line 27 - genes controlling cholesterol...
- Line 30 - PPN shortcut should be where cited for the first time (line 23)
- Line 48 - control chromatin
- Line 50 - done - discovered
- Line 56 - influence these
- Line 58 - amino acids content/addition, as reduction of boar taint
- I think authors should not use comma before and
- Line 70 - that cannot
- Line 73 - and are necessary to control
- Line 74 - ore in some cases produced synthetically
- Line 86 - animal what is/as interaction
- Lines 90, 135 - genetic breed
- Line 92 - parameter that reflects feed intake
- Line 110 - performed in some
- Line 131 - shown
- Figure 1. - should be bigger or with higher resolution because it is illegible
- Line 163 - catalyzed by three...in mammals - DNA methylotransferases (DNMTs) - DNMT1, DNMT3a, DNMT3b
- Genes symbols should be italicized in whole text for example lines 178, 179, 181, 184, 322, 350, 365, please check the others
- Line 88 - important
- Line 197 - has been found in pigs for most
- Line 231 - necessary
- Line 235 - the most studied nutrients
- Line 236-7 - modifies ... modification - replace one
- Line 249 - Latin species name should be italicized
- Figure 4. with its description in text and below figure - standardize the names for example XPOR5 vs EXPO5, DGCR8 vs DGR8
- Lines 271-2 - embryos are not tissue similar to nervous system - better use nervous tissue
- Line 285 - casein protein, gluten protein
- Line 286 - change gene levels? maybe expression...
- Line 289 - gene expression of serine dehydratase in liver
- Line 291- prenancy - pregnant
- Line 305 - i think skeletal muscle and liver
- Line 311 - of
- Line 313 - decrease
- Line 323 - expression of IGF2 mRNA
- Line 325 - sows
- Line 224 - are used to
- Line 335 - separate sentence (too long) - ...requirements. All of them..
- Line 350 - IGF2 gene affects gluconeogenesis (line 349)
- Line 358 - both in female than piglets - it is not clear
- Line 359 - increase
- Line 377 - acetylation and histone
- Line 279 - posttraductional, in the Reference [124] is post-translational
- Lines 388,391 - posttraductional, in the Reference [129] is post-transcriptional
- Line 382 - feed-to-gain conversion ratio
- Line 384 - effects are unknown
- Lines 385-6 - genes that code for amino acid transporters and muscle growth, as
- Line 405 - analyzed ... levels in diet on
ENGLISH LANGUAGE MUST BE CHECK AND CORRECT!!!
Author Response
Dear reviewer,
The authors would like to thank you for the useful feedback on our review.
We have made several editorial changes in response to the critiques and have included these in the revised manuscript. The English language has been checked and corrected.
The following is a point-by-point response with line numbers added to indicate where specific changes have been made. We have included the reviewer comments and requests in black font followed by our responses in blue font.
- sentences are too long and sometimes hard to understand
Following the reviewer's recommendations, the manuscript has been revised and some sentences have been rewritten.
- line 21 – tried
Done.
- line 27 - genes controlling cholesterol...
Done.
- Line 30 - PPN shortcut should be where cited for the first time (line 23)
Done.
- Line 48 - control chromatin
Done.
- Line 50 - done – discovered
Done.
- Line 56 - influence these
Done.
- Line 58 - amino acids content/addition, as reduction of boar taint
Done.
- I think authors should not use comma before and
The commas has been added.
- Line 70 - that cannot
“it” has been removed.
- Line 73 - and are necessary to control
Done.
- Line 74 - ore in some cases produced synthetically
Done.
- Line 86 - animal what is/as interaction
The sentence has been rewritten as “For the correct evaluation of the epigenetic effects of some nutrients, it is necessary to know the quality and quantity of them, and their interaction.” to facilitate its understand.
- Lines 90, 135 - genetic breed
“Genetic” has been removed.
- Line 92 - parameter that reflects feed intake
Done.
- Line 110 - performed in some
Done.
- Line 131 – shown
Done.
- Figure 1. - should be bigger or with higher resolution because it is illegible
Figure 1 has been modified with high quality and the references has been added.
- Line 163 - catalyzed by three...in mammals - DNA methylotransferases (DNMTs) - DNMT1, DNMT3a, DNMT3b
Done.
- Genes symbols should be italicized in whole text for example lines 178, 179, 181, 184, 322, 350, 365, please check the others
Done.
- Line 88 – important
Done.
- Line 197 - has been found in pigs for most
Done.
- Line 231 – necessary
Done.
- Line 235 - the most studied nutrients
Done.
- Line 236-7 - modifies ... modification - replace one
“modification” has been removed.
- Line 249 - Latin species name should be italicized
Done.
- Figure 4. with its description in text and below figure - standardize the names for example XPOR5 vs EXPO5, DGCR8 vs DGR8
Done.
- Lines 271-2 - embryos are not tissue similar to nervous system - better use nervous tissue
Done.
- Line 285 - casein protein, gluten protein
Done.
- Line 286 - change gene levels? maybe expression...
Done.
- Line 289 - gene expression of serine dehydratase in liver
Done.
- Line 291- prenancy – pregnant
Done.
- Line 305 - i think skeletal muscle and liver
Done.
- Line 311 – of
Done.
- Line 313 – decrease
Done.
- Line 323 - expression of IGF2 mRNA
Done.
- Line 325 – shows
Done.
- Line 224 - are used to
Done.
- Line 335 - separate sentence (too long) - ...requirements. All of them..
Done.
- Line 350 - IGF2 gene affects gluconeogenesis (line 349)
Done.
- Line 358 - both in female than piglets - it is not clear
The sentence has been rewritten as “metabolism in females and piglets”.
- Line 359 – increase
Done.
- Line 377 - acetylation and histone
Done.
- Line 279 - posttraductional, in the Reference [124] is post-translational
Done.
- Lines 388,391 - posttraductional, in the Reference [129] is post-transcriptional
Done.
- Line 382 - feed-to-gain conversion ratio
Done.
- Line 384 - effects are unknown
Done.
- Lines 385-6 - genes that code for amino acid transporters and muscle growth, as
Done.
- Line 405 - analyzed ... levels in diet on
Done.

Round 2
Reviewer 1 Report
Dear Authors,
Thank you for your response to my comments. Although, most of my comments have been addressed. The authors should add some text to let the reader know about the scarce information for other ncRNAs as they state "The authors fully agree with the reviewer's comment about other types of epigenetic regulation involving other RNAs, such as ncRNAs or lcn-RNAs. However, the bibliography about the relationship between epigenetic regulation by these RNA molecules and protein nutrition is very scarce, so the authors decided not to include them in the manuscript."
The authors should make a table to summarize all epigenetic changes together with the references.
Author Response
Dear reviewer,
Thank you very much for your comments. Following your suggestion, a section "other non-coding RNAs as epigenetic regulators" has been added. Further, Table 1 that summarizes different non-coding RNAs related to epigenetics, their function and references has been also added.
